# Seroprevalence of COVID-19 infection in a rural district of South India: A population-based seroepidemiological study

Leeberk Raja Inbaraj[1], Carolin Elizabeth George[1]*, Sindhulina Chandrasingh[2]

**1** Division of Community Health and Family Medicine, Bangalore Baptist Hospital, Bangalore, Karnataka, India, **2** Department of Microbiology, Bangalore Baptist Hospital, Bangalore, Karnataka, India

* carolinelizabethj@gmail.com

**Data Availability Statement:** Data has been attached as a Supporting Information file (S1 Data).

## Abstract

### Objectives

We aimed to estimate the seroprevalence of COVID-19 in a rural district of South India, six months after the index case.

### Methodology

We conducted a cross-sectional study of 509 adults aged more than 18 years. From all the four subdistricts, two grampanchayats (administrative cluster of 5–8 villages) were randomly selected followed by one village through convenience. The participants were invited for the study to the community-based study kiosk set up in all the eight villages through village health committees. We collected socio-demographic characteristics and symptoms using a mobile application-based questionnaire, and we tested samples for the presence of IgG antibodies for SARS CoV-2 using an electro chemiluminescent immunoassay. We calculated age-gender adjusted and test performance adjusted seroprevalence.

### Results

The age-and gender-adjusted seroprevalence was 8.5% (95% CI 6.9%- 10.8%). The unadjusted seroprevalence among participants with hypertension and diabetes was 16.3% (95% CI:9.2–25.8) and 10.7% (95% CI: 5.5–18.3) respectively. When we adjusted for the test performance, the seroprevalence was 6.1% (95% CI 4.02–8.17). The study estimated 7 (95% CI 1:4.5–1:9) undetected infected individuals for every RT-PCR confirmed case. Infection Fatality Rate (IFR) was calculated as 12.38 per 10000 infections as on 22 October 2020. History of self-reported symptoms and education were significantly associated with positive status (p < 0.05)

### Conclusion

A significant proportion of the rural population in a district of south India remains susceptible to COVID-19. A higher proportion of susceptible, relatively higher IFR and a poor tertiary healthcare network stress the importance of sustaining the public health measures and

**Funding:** Authors were supported by Azim Premji Foundation (https://azimpremjifoundation.org/) This was not a research grant but a philanthropic support for COVID response. The funders had no role in study design, data collection and analysis, decision to publish, or preparation of the manuscript.

**Competing interests:** The authors have declared that no competing interests exist.

promoting early access to the vaccine are crucial to preserving the health of this population. Low population density, good housing, adequate ventilation, limited urbanisation combined with public, private and local health leadership are critical components of curbing future respiratory pandemics.

## Introduction

Coronavirus disease (COVID-19) was declared as a global pandemic by the World Health Organization on 11 March 2020 [1]. Globally, more than 60 million confirmed cases of COVID-19, including 1,416,292 deaths, have been reported to WHO as of 26 November 2020 [2]. India has reported more than 9.2 million cases with more than with 135,223 deaths and Karnataka- a south Indian state had 894,137 cases with 8,512 deaths as of 26 November 2020 [3, 4].

There has been substantial evidence that a large proportion of the people infected with SARS CoV-2 are asymptomatic, but they can infect others. It has been reported based on an analysis of 21 published reports that asymptomatic cases could account from 5 to 80% [5]. It is crucial to recognise an infected person early and break the route of transmission to control COVID-19. However, in reality, they do not require or seek medical attention and contribute to the rapid spread of the disease [6]. Hence, health authorities cannot totally rely on confirmed cases of COVID-19 detected by RT-PCR as it could potentially miss asymptomatic and pre-symptomatic infections for containment measures. In order to overcome this challenge, WHO and others have recommended population-based seroepidemiological studies to generate data and to implement containment measures accordingly [7]. These surveys also can give us an estimation of the proportion of the population still susceptible to the infection as it is assumed that antibodies provide immunity.

Indian Council of Medical Research (ICMR) has conducted a nationwide serosurvey among 21 states and reported a population-weighted seroprevalence of 0.73% between May and June 2020 [8]. While a hospital-based survey from Srinagar, northern India has estimated gender-standardised seroprevalence of 3.6% in July 2020 and our study from one of the largest slums in Bangalore revealed a seroprevalence of 57% in September 2020 [9, 10].

Community Health Division (CHD) of Bangalore Baptist Hospital has been providing curative and preventive health services through a Rural Health centre and network of mobile clinics to residents of Bangalore rural district over a decade. CHD also runs special programs for chronic diseases, disability rehabilitation and alcohol de-addiction. One of our flagship programs is home-based rural palliative care program which has benefited numerous patients with terminal illness ever since it was initiated in 2005. Our grass root workers continued to do home visits to provide home care, monitor blood pressure and blood sugar, and to educate the community about COVID-19. However, we had stopped our mobile clinics to reduce the urban-rural transmission of infection. As there can be considerable variation in the seroprevalence based on geographical setting and density of the population, knowledge of seroprevalence in this community help us to conduct a risk-benefit analysis of certain services like mobile clinics, which improves access to medical care at the cost of spreading the virus to the rural community. Hence, we designed a community-based cross-sectional study to estimate a seroprevalence in Bangalore rural district six months after the index case. We also hope the findings of this study will help the health authorities in disease containment and add valuable data to researchers across the globe.

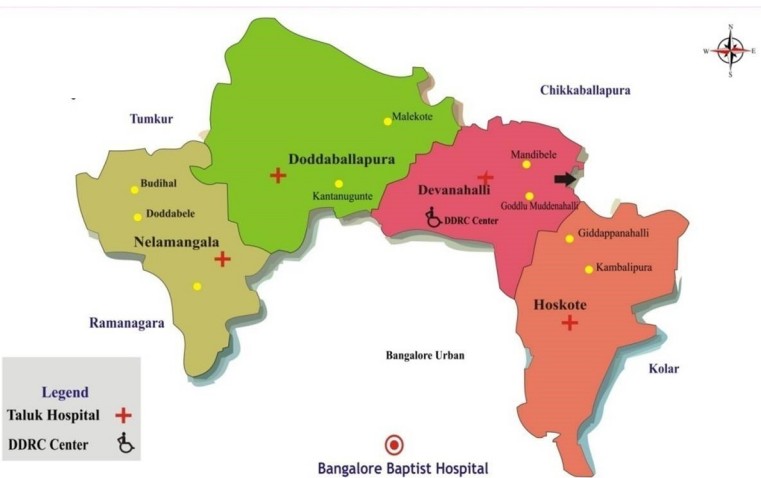

**Fig 1. Pictorial map of Bangalore rural district.**

## Materials and methods

### Setting

The study was conducted in Bangalore Rural District of Karnataka, a South Indian State. This district is located close to Bangalore city and is divided into four sub-districts (taluks) and 105-gram panchayats which are village administrative units (Fig 1). Each gram panchayat has a cluster of 10–15 villages [11]. According to the Indian Census 2011, the population was 9,90,923, and the sex-ratio was 945 females for every male, which is lesser than the state of the average of 973. The district has a population density of 441 inhabitants per square kilometre. The villagers are dependent on farming and farming related activities such as cattle rearing for their income. However, the advent of Special Economic Zone (SEZ) and Bangalore International Airport in this district, service and Information technology industries are also booming in the past few years [12]. Health care in the district is delivered through the Government health system and private practitioners. Apart from primary health care, BBH also established strong community connections through the formation of village health committees and linking with the rural self-government.

### Study design and sample size

We designed a cross-sectional seroepidemiological survey in Bangalore rural district based on the recommendation of WHO as the most appropriate study design [13]. A serosurvey from a densely populated slum in Mumbai, India reported a seroprevalence 57%, and we assumed a seroprevalence of 30% (lower risk of transmission in rural areas compared to slums) in the rural district. We calculated a minimum sample size of 504 with 5% absolute and a design effect of 1.5 [14].

### Data collection

Bangalore Rural District consists of four sub-districts which are further divided into grampanchayats which are village administrative units. From the four sub-districts, we randomly chose two grampanchayats. In each grampanchayat, a village which is centrally located was chosen as a sampling unit based on convenience, and a kiosk was set up in one of the trusted community spaces in each village. Our community health workers invited people (adults ≥18years) from

houses to give blood samples. If a household refused to participate, then the next house was approached. In each cluster, mobilisation continued till the desired sample size was achieved in each village. We aimed to include 63 adults from each of these villages, adding up to 126 in each sub-district.

We met with the village health committees and discussed the purpose of the study and enlisted their cooperation. Together with the community, we decided that kiosk-based recruitment of the participants was more practical due to strict restriction on the movement of the people by the state government. People were also apprehensive about the health team from cities visiting their homes and increasing the risk of transmission of the disease.

We recruited people after explaining the purpose of the study, took written consent and then interviewed people with a questionnaire by a trained research coordinator who had previous training in data collection. An Epi-info 7.0 $^{TM}$ mobile application-based tool was used to capture responses offline by the interviewer, and it was later downloaded for analysis.

The questionnaire contained questions about demographic information (age, gender, education, comorbidities such as diabetes, hypertension, lung disease and cancer), history of exposure to COVID-19 infection (history of being diagnosed as COVID-19 case, interaction and household contacts with persons with confirmed COVID-19), any history of COVID-19 related symptoms a month before the survey. After the completion of the interview, our phlebotomists collected 5ml of blood from each participant via venepuncture in a plain vacutainer. They transported it to BBH laboratory within 5 hours, maintaining the cold chain.

## Sample processing and analysis

The serum was separated and used to test for antibodies using the Elecsys Anti SARS CoV2 assay, an electro chemiluminescent immunoassay using a recombinant protein representing the nucleocapsid (N) antigen for the determination of high-affinity antibodies (including IgG) against SARS CoV2 [15]. This assay employs a cut-off index (COI) that is automatically calculated from two calibration standards—a COI of 1.0 or more is considered reactive/positive, and a COI less than or equal to 1.0 is reported as nonreactive/negative. The assay sensitivity and specificity were reported to be 97·2% (95·4–98·4) and 99·8% (99·3–100) respectively, in samples taken 30 days or more post symptom onset [16]. A unique identification number was used to link the interview information and laboratory results.

## Statistical analysis

We used Statistical Package for the Social Sciences version 20.0 and STATA version 15.0 for statistical analysis. The frequency of characteristics of the survey participants was described. Unadjusted seroprevalence of COVID-19 IgG antibody was reported in per cent with 95% confidence interval (CI). We used rural area figures of Karnataka from the Sample Registration System (SRS) statistical report 2018 to calculate weights for reporting age-and-gender standardised seroprevalence [17] (S1 Table).

Case-to-undetected-infections ratio (CIR), was calculated as a ratio of the number of reported RT-qPCR-confirmed COVID-19 cases two weeks before the imitation of serosurvey to the number of people who have antibodies in our study. This was based on an earlier study reported median seroconversion times for total antibodies, IgM and then IgG at day-11, day-12 and day-14, respectively based on hospitalised patients and seroconversion for IgG and IgM is reported to occur simultaneously or sequentially [18, 19]. Assuming a three-week lag time from infection to death, we considered the reported number of fatalities after three weeks of the survey to estimate the plausible range of the infection fatality ratio (IFR) [16]. It was calculated as the number of deaths reported upon the total number of people with high-affinity

antibodies per 10000 infections. We used a projected population of 2020 Bangalore rural district using 2011 census data prepared by Directorate of Economics and statistics, Bangalore 2013 to calculate all these parameters [20]. The association of seroprevalence with comorbid conditions and socio-demographic characteristics was tested using chi-square tests.

### Ethical consideration

The Ethics Committee on Bangalore Baptist Hospital approved the survey protocol on 30 June 2020. Written informed consent was obtained from the participants, and the test results were communicated to them.

### Results

Our 509 participants were almost equally distributed in four taluks (Doddaballapur– 28.6%, Devanahalli -23.5%, Nelamangala– 25.7%, Hoskote -20.2%) of Bangalore rural district. The mean age was 47.0 +/ 16.4 years, and the majority were men (52.3%). Most people (86.0%) had less than 12 years of education, and 28.4% were either not working or homemakers. Hypertension (20.2%) and diabetes (16.9%) were reported as the most common comorbidities. (Table 1). Among 509 individuals, 7/509 (1.37%) had at least one symptom suggestive of COVID-19 in the last three months before the interview, 5/509 (0.98%) reported a history of an infected family member, and none of them gave a history exposure to an infected person in the past or tested positive for COVID-19

The overall seroprevalence of COVID-19 was 12.4% (95% CI 9.6–15.6) (Table 2) The age- and gender-adjusted seroprevalence was 8.5% (95% CI 6.9%- 10.8%) (S1 Table). The unadjusted seroprevalence among participants with hypertension and diabetes was 16.3% (95% CI:9.2–25.8) and 10.7% (95% CI: 5.5–18.3) respectively, but the association with seropositivity was not significant. Among seropositive individuals, one participant reported a history of a

**Table 1. Socio-demographic characteristics of the study population.**

| Demographics | Variables | N | Percentage |
|---|---|---|---|
| Age Group | ≤20 | 18 | 3.5 |
| | 21–40 | 177 | 34.8 |
| | 41–60 | 208 | 40.9 |
| | >60 | 106 | 20.8 |
| Sex | Male | 266 | 52.3 |
| | Female | 243 | 47.7 |
| Education | Illiterate | 125 | 24.6 |
| | Primary | 96 | 18.9 |
| | Middle school | 05 | 1.0 |
| | High School | 117 | 23.0 |
| | PUC/Diploma | 95 | 18.7 |
| | Degree | 71 | 13.9 |
| Occupation | Housewife | 58 | 11.4 |
| | Domestic Helper | 12 | 2.4 |
| | Daily wage labourer | 49 | 9.6 |
| | Notworking | 87 | 17.1 |
| | Professional | 81 | 15.9 |
| | Others | 61 | 12.0 |
| Comorbidities | Diabetes | 103 | 20.2 |
| | Hypertension | 86 | 16.9 |

**Table 2. Unadjusted seroprevalence of COVID-19 in Bangalore rural district, India.**

|  | Category | Male | Prevalence (95% CI) | Female | Prevalence (95% CI) | Total | Overall prevalence (95% CI) |
|---|---|---|---|---|---|---|---|
| Age (yrs) | ≤20 | 10 | 10 (0.3–44.5) | 8 | 0 | 18 | 5.6 (0.1–27.3) |
|  | 21–40 | 86 | 10.5 (4.9–18.9) | 91 | 16.5 (9.5–25.7) | 177 | 13.6 (8.9–19.5) |
|  | 41–60 | 120 | 13.3 (7.8–20.7) | 88 | 11.4(5.6–19.9) | 208 | 12.5 (8.3–17.8) |
|  | >60 | 50 | 12.0 (4.5–24.3) | 56 | 10.7 (4.0–21.9) | 106 | 11.3 (6.0–18.9) |
|  | Total | 266 | 12.0 (8.4–16.6) | 243 | 12.0 (8.8–17.6) |  | 12.4 (9.6–15.6) |

family member being positive in the last three months before the interview. The majority (92.0%) of the seropositive individuals, did not report any symptom related to COVID-19 infection at the time of the study nor in the past.

We estimated that the cumulative number of SARS CoV-2 (Severe acute respiratory syndrome coronavirus 2) infection in Bangalore rural district was 96,874 (95% CI 78638–123086) during two weeks before the beginning of the study (17 September to 1 October 2020). When we adjusted for sensitivity and specificity of the test kit, the seroprevalence was 6.1% (95% CI 4.02–8.17) and the cumulative number of infections was 69,521 (95% CI 45815–92315).

The cumulative number of RT-PCR confirmed cases till 2 October was 100,54 in Bangalore rural district. The study estimated 7 (96,854/100,54) undetected infected individuals for every RT-PCR (Reverse Transcription Polymerase Chain Reaction) confirmed case, i.e., case-to-undetected-infections ratio (CIR) of 1:7 and CIR could range from 1:4.5 to 1:9. Based on age-gender adjusted seroprevalence rate, the Infection Fatality Rate (IFR) was calculated as 12.38 per 100,00 infections as on 22 October 2020 in Bangalore rural district.

Age, gender, occupation, presence of comorbidities were not associated with positive status (p-value >0.05). In contrast, the history of at least one self-reported symptoms suggestive of COVID-19 in the last three months before the study (71.4% Vs 11.6%) and higher education (15.6% Vs 8.4%) were significantly associated with seropositivity (Table 3).

**Table 3. Factors associated with seropositivity of COVID-19.**

| Factors | Categories | Serological status | | Total | p value |
|---|---|---|---|---|---|
|  |  | Reactive | Non- Reactive |  |  |
| Age in years | < = 40 | 25 (12.8) | 170 (87.2) | 195 | 0.81 |
|  | >40 | 38 (12.1) | 276 (87.9) | 314 |  |
| Gender | Male | 32 (12) | 234 (88) | 266 | 0.8 |
|  | Female | 31 (12.8) | 212 (87.2) | 243 |  |
| Education | Lower (< = 8 years) | 19 (8.4) | 207 (91.6) | 226 | 0.01* |
|  | Higher (>8 years) | 44 (15.6) | 239 (84.4) | 263 |  |
| Occupation | Farmer/Daily wage labour | 21 (10) | 189 (90) | 210 | 0.71 |
|  | Others | 42 (14) | 257 (86) | 299 |  |
| Hypertension | Yes | 14 (16.3) | 72 (83.7) | 86 | 0.22 |
|  | No | 49 (11.6) | 374 (88.4) | 423 |  |
| Diabetes | Yes | 11 (10.7) | 92 (89.3) | 103 | 0.5 |
|  | No | 52 (12.8) | 354 (87.2) | 406 |  |
| No. of rooms in the house | < = 2 | 63 (12.6) | 438 (87.4) | 501 | 0.28 |
|  | >2 | 0 | 8 (100) | 8 |  |
| History of at least one symptom suggestive of COVID-19 | Yes | 5 (71.4) | 2 (28.6) | 7 | <0.001* |
|  | No | 58 (11.6) | 444 (88.4) | 502 |  |

*significant p value.

## Discussion

Our study revealed that a large proportion of the rural population remains susceptible to infection and far from reaching the seroprevalence required for herd immunity. A serosurvey from Karnataka during the same period (3–16 September) has reported a sightly higher seroprevalence of 15.2% in the same district [21]. The variation can be attributed to samples collected from multiple settings, including hospital settings and among the high-risk group.

The low prevalence in this rural district is in contrast to an earlier study by the same investigators that was conducted in a dense urban slum in Bangalore City, which showed a seroprevalence of 57% [9]. This is expected as the participants in the current study, lived in sparsely populated villages, in well-ventilated houses and with the privilege of less polluted air. When the government-imposed lockdown, the adequacy of minimal resources for sustenance and the self-reliance of the villages, reduced travel to cities a minimum. These factors would have limited the spread of the infection.

Our study estimated that there were 5 to 9 undetected infected individuals for every RT-PCR confirmed case. This shows that most of the infections were picked up the existing testing infrastructure. CIR in Bangalore rural district was close to reported estimates of Bangalore city (1.10), can be accounted to its proximity to the city [21]. CIR reported in this study, is much lower than other studies in the western countries, probably due to high testing rate in India during recent months [8, 18]. However, slums from Bangalore city reported a high CIR of 1:195 as compared to rural counterparts. Poor health infrastructure and high prevalence of stigma leading to underreporting may be the reason for high CIR in slums [9].

We found age and comorbidities were not significantly associated with seropositivity. Though advanced age and comorbidities are associated with severe illness, there is limited data regarding increased COVID-19 susceptibility with mild asymptomatic cases [22]. The hospital-based study from Srinagar found that people between the age group of 30–69 years had higher odds of being seropositive (IgG) as compared to the younger population, but they did not find any gender difference in seropositivity [10]. However, the nationwide survey showed male gender was significantly associated with seropositivity than females [8]. Age and gender have a profound influence on mobility and is varied across cultures. Hence the susceptibility to infection can be attributed to the function of mobility rather than age and gender per se.

Though diabetes has been associated with increased mortality in COVID-19, the susceptibility to the infection may be same as the general population [23]. The same was reflected in our study. Though rural, this population had access to diagnosis and treatment of common comorbidities due to the outreach of the public-spirited hospital and the government health system. Access to chronic medications was facilitated even during the lockdown and intense resource reallocation following COVID-19, through our grass-root health workers who delivered medicines at home for people with chronic diseases to keep their diseases under control. We could imply that the efforts to sensitise the public regarding COVID-19 by the government and private sectors in sensitisation have played a valuable role.

We estimated an IFR of 12.8 per 10000 infections or 0.13%, which is comparable to what is reported from the Indian subcontinent (0.27–1.03) and other countries like USA (0.12–0.2%) Iran (0.08–0.12%), Brazil and Spain (1%) [8, 21, 24]. Estimating IFR is a challenge as it will depend on infection rate (seroprevalence) and the robustness of system capturing mortality. Both variables have estimation challenges of varying degrees in different parts of India. Since there are only a handful of studies estimating seroprevalence, we have only limited studies to compare. IFR reported in a study conducted in a Bangalore slum during the same period was absurdly low (0.03%), which can be attributed to under-reporting of deaths rather than reduced fatality in urban slums [9].

We could draw several implications from the findings of the study. First and foremost, rural areas succeeded in halting the spread of infection to a greater extent as compared to cities. However, rural areas are challenged by the poor health system and low cash economy, distancing itself from urbanisation reaped overall health benefits to people in villages, in terms of the number of infections. This is a reminder that guarded urbanisation preserving the natural ecosystem is an essential determinant of health.

Secondly, strict containment strategies like lockdown curbed infection without profound livelihood implication in this rural setting. This was possible because of the strength of the local economy and reduced inequities. The villages had enough seasonal grains (ragi, a millet), home-grown vegetables and dairy products for nourishment. Since the population density was less, there was enough water for the increased demand for handwashing, clean air to breathe, and physical distancing was a practical possibility. Strong social connections, a powerful rural disposition added value during COVID-19 infections. Neighbourhoods took care of infected households with food and essential medicines and arranged for a referral if they need hospital support.

Thirdly, low seroprevalence should be looked in two ways. One way to look at this 'achievement', success in preventing the spread and the other way to look at it as 'responsibility' due to susceptibility. Since we assume that other villages in India have similar or a slightly lesser seroprevalence, we need to keep in mind' huge susceptible burden' as 68.84% of India's population live in villages according to the census (2011) [25]. This has potential to staggering peaks and gives a warning signal for policymakers about the possibilities of multiple waves of the pandemic. In this context, discussion on sustaining safety measures and access to vaccination is of paramount importance.

The study has potential biases. Though all the subdistricts were selected, and subsequently villages were randomly selected, we employed convenience sampling at the village level. Villages were apprehensive about the medical team from the city, and hence we enrolled based on individual preference. This would have resulted in selection bias; however, we tried to reduce the bias by calculating age-gender adjusted seroprevalence. Another possibility is the occurrence of measurement bias in estimating seroprevalence. Since we have not done RT-PCR, we would have missed the current infection and underestimated the prevalence. Measurement bias can also be due to validity parameters of the test, which we have addressed through test performance adjusted seroprevalence.

There are many strengths to this study. This is one of the earliest population-based seroprevalence study conducted in a rural district of India harbouring a million people. This contributes to the body of evidence regarding the virus, its spread and the future implications in the rural context. The study being conducted by researchers who knows the population closely is an added advantage as the results are discussed in relation to the contextual realities.

This study has a few limitations. We did not follow a strict probability sampling technique due to feasibility reasons. Another limitation is that we did not estimate the current infection using RT-PCR. Both these aspects have an effect on the true estimation of seroprevalence in this community. Though a 15 days recall period is generally recommended for eliciting morbidity, we have used a longer (3 months) recall. Our assumption was that people would recall their symptoms related to COVID for a longer period due to the unusually significant nature of this pandemic and the attention it had received from media. However, this could have resulted in recall bias. Another limitation is that we have limited our research to one rural district; hence the generalisation of the findings has to be done with caution.

## Conclusion

We found a low seroprevalence of COVID-19 infection among rural population in a district of South India, six months after the index case. The age-old public health measures of low

population density, good housing, adequate ventilation, hygiene measures combined with the public, private and local health leadership limited the spread of an infectious respiratory viral pathogen in this low resource setting. Since more than three fourth of the rural population remains susceptible to COVID-19, sustaining public health measures and promoting access to vaccination is of utmost importance to safeguard the health this population as severe COVID-19 can be overtly burdensome owing to poor tertiary healthcare landscape of the rural setting.

## Supporting information

**S1 Table. Age- and gender-standardized seroprevalence.**
(LOG)

**S1 Data. Data.**
(SAV)

## Acknowledgments

We would like to thank Mr Tata Rao creating Epi-data for data collection, conducting preliminary analysis and his help with referencing, and data collection. We also acknowledge the support of Ms Sangeetha and Mr Prakash in their assistance in data collection. We would like to extend our thanks Dr Arun, Epidemiologist from Christian Medical College, Vellore for his help during statistical analysis. We are grateful for the support of people Bangalore rural district for their participation in the study.

## Author Contributions

**Conceptualization:** Leeberk Raja Inbaraj, Carolin Elizabeth George, Sindhulina Chandrasingh.

**Data curation:** Leeberk Raja Inbaraj.

**Formal analysis:** Leeberk Raja Inbaraj, Carolin Elizabeth George.

**Funding acquisition:** Carolin Elizabeth George.

**Investigation:** Leeberk Raja Inbaraj, Carolin Elizabeth George, Sindhulina Chandrasingh.

**Methodology:** Leeberk Raja Inbaraj, Carolin Elizabeth George, Sindhulina Chandrasingh.

**Project administration:** Leeberk Raja Inbaraj, Carolin Elizabeth George.

**Resources:** Carolin Elizabeth George.

**Software:** Leeberk Raja Inbaraj.

**Supervision:** Sindhulina Chandrasingh.

**Validation:** Leeberk Raja Inbaraj, Carolin Elizabeth George, Sindhulina Chandrasingh.

**Writing – original draft:** Leeberk Raja Inbaraj.

**Writing – review & editing:** Carolin Elizabeth George, Sindhulina Chandrasingh.

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
