## [Decision Letter · Decision Letter 0]

8 Mar 2021

PONE-D-20-39600

Seroprevalence of COVID-19 infection in a rural district of South India: a community-based cross-sectional study.

PLOS ONE

Dear Dr. Carolina George,

Thank you for submitting your manuscript to PLOS ONE. After careful consideration, we feel that it has merit but does not fully meet PLOS ONE’s publication criteria as it currently stands. Therefore, we invite you to submit a revised version of the manuscript that addresses the points raised during the review process.

We look forward to receiving your revised manuscript.

Kind regards,

Muralidhar M. Kulkarni

Academic Editor

PLOS ONE

Journal Requirements:

3. We note that Figure 1 in your submission contains map images which may be copyrighted. All PLOS content is published under the Creative Commons Attribution License (CC BY 4.0), which means that the manuscript, images, and Supporting Information files will be freely available online, and any third party is permitted to access, download, copy, distribute, and use these materials in any way, even commercially, with proper attribution. For these reasons, we cannot publish previously copyrighted maps or satellite images created using proprietary data, such as Google software (Google Maps, Street View, and Earth). For more information, see our copyright guidelines: http://journals.plos.org/plosone/s/licenses-and-copyright.

(1) You may seek permission from the original copyright holder of Figure 1 to publish the content specifically under the CC BY 4.0 license. 

Additional Editor Comments:

Dear Dr. Carolina,

This is to inform you that there are some minor revisions required for your manuscript. We request you to please do it and submit it for review. We are giving full consideration in accepting your manuscript and the same will be communicated to you after satisfactory address of the comments.

Reviewers' comments:

Reviewer's Responses to Questions

**Comments to the Author**

1. Is the manuscript technically sound, and do the data support the conclusions?

Reviewer #1: Yes

2. Has the statistical analysis been performed appropriately and rigorously? 

Reviewer #1: Yes

3. Have the authors made all data underlying the findings in their manuscript fully available?

Reviewer #1: Yes

4. Is the manuscript presented in an intelligible fashion and written in standard English?

Reviewer #1: Yes

5. Review Comments to the Author

Reviewer #1: The manuscript is well written except for minor corrections and clarifications

The results of association between different variables and seropositivity can be presented as a table

what is the justification for selecting three months for symptom recall?

There are minor typos throughout the manuscript that need to be fixed

6. PLOS authors have the option to publish the peer review history of their article (what does this mean?). If published, this will include your full peer review and any attached files.

Reviewer #1: **Yes: **Dr Sneha Deepak Mallya

---

## [Author Response · Author response to Decision Letter 0]

10 Mar 2021

We thank the editor and reviewer for taking the time and going through our manuscript (PONE-D-20-39600). We appreciate the constructive feedback received to improve the quality of the paper. We have addressed the points raised and submitted a revised version of the manuscript with highlights and given our response in the attached document.

---

## [Decision Letter · Decision Letter 1]

15 Mar 2021

Seroprevalence of COVID-19 infection in a rural district of South India: a population-based seroepidemiological study

PONE-D-20-39600R1

Dear Dr. George,

We’re pleased to inform you that your manuscript has been judged scientifically suitable for publication and will be formally accepted for publication once it meets all outstanding technical requirements.

Kind regards,

Muralidhar M. Kulkarni

Academic Editor

PLOS ONE

Additional Editor Comments (optional):

Reviewers' comments:

Reviewer's Responses to Questions

**Comments to the Author**

1. If the authors have adequately addressed your comments raised in a previous round of review and you feel that this manuscript is now acceptable for publication, you may indicate that here to bypass the “Comments to the Author” section, enter your conflict of interest statement in the “Confidential to Editor” section, and submit your "Accept" recommendation.

Reviewer #1: All comments have been addressed

2. Is the manuscript technically sound, and do the data support the conclusions?

Reviewer #1: Yes

3. Has the statistical analysis been performed appropriately and rigorously? 

Reviewer #1: Yes

4. Have the authors made all data underlying the findings in their manuscript fully available?

Reviewer #1: Yes

5. Is the manuscript presented in an intelligible fashion and written in standard English?

Reviewer #1: Yes

6. Review Comments to the Author

Reviewer #1: p value in table 3 to be corrected. All corrections are satisfactorily addressed

The paper is well written and contributes significantly to current evidence .

7. PLOS authors have the option to publish the peer review history of their article (what does this mean?). If published, this will include your full peer review and any attached files.

Reviewer #1: **Yes: **Sneha Deepak Mallya

---

## [Editor Report · Acceptance letter]

22 Mar 2021

PONE-D-20-39600R1 

Seroprevalence of COVID-19 infection in a rural district of South India: a population-based seroepidemiological study 

Dear Dr. George:

I'm pleased to inform you that your manuscript has been deemed suitable for publication in PLOS ONE. Congratulations! Your manuscript is now with our production department. 

Kind regards, 

on behalf of

Dr. Muralidhar M. Kulkarni 

Academic Editor

PLOS ONE